energy/environmental engineering/
environmental chemistry

rainwater, microbial fuel cell, remote area sensor, power generation

**Author for correspondence:**
Nasser A. M. Barakat
e-mail: nasbarakat@minia.edu.eg

# Rainwater-driven microbial fuel cells for power generation in remote areas

Mohamed Taha Amen[1,4], Ahmed S. Yasin[1], Mohamed I. Hegazy[4], Mohammad Abu Hena Mostafa Jamal[3], Seong-Tshool Hong[3] and Nasser A. M. Barakat[2]

[1]Bio-Nanosystem Engineering Department, Chonbuk National University, Jeonju 561-756, Republic of South Korea
[2]Chemical Engineering Department, Faculty of Engineering, Minia University, Minia, Egypt
[3]Department of Biomedical Sciences and Institute for Medical Science, Chonbuk National University Medical School, Jeonju, Chonbuk, Korea
[4]Microbiology Department, Faculty of Agriculture, Zagazig University, Zagazig, Egypt

MTA, 0000-0002-3623-1359

The possibility of using rainwater as a sustainable anolyte in an air-cathode microbial fuel cell (MFC) is investigated in this study. The results indicate that the proposed MFC can work within a wide temperature range (from 0 to 30°C) and under aerobic or anaerobic conditions. However, the rainwater season has a distinct impact. Under anaerobic conditions, the summer rainwater achieves a promised open circuit potential (OCP) of $553 \pm 2$ mV without addition of nutrients at the ambient temperature, while addition of nutrients leads to an increase in the cell voltage to $763 \pm 3$ and $588 \pm 2$ mV at 30°C and ambient temperature, respectively. The maximum OCP for the winter rainwater ($492 \pm 1.5$ mV) is obtained when the reactor is exposed to the air (aerobic conditions) at ambient temperature. Furthermore, the winter rainwater MFC generates a maximum power output of $7 \pm 0.1$ mWm$^{-2}$ at a corresponding current density value of $44 \pm 0.7$ mAm$^{-2}$ at 30°C. While, at the ambient temperature, the maximum output power is obtained with the summer rainwater ($7.2 \pm 0.1$ mWm$^{-2}$ at $26 \pm 0.5$ mAm$^{-2}$). Moreover, investigation of the bacterial diversity indicates that *Lactobacillus* spp. is the dominant electroactive genus in the summer rainwater, while in the winter rainwater, *Staphylococcus* spp. is the main electroactive bacteria. The cyclic voltammetry analysis confirms that the electrons are delivered directly from the bacterial biofilm to the anode surface and without mediators. Overall, this study opens a new avenue for using a novel sustainable type of MFC derived from rainwater.

# 1. Introduction

A microbial fuel cell (MFC) is a system in which the microorganisms can convert the chemical energy embedded in some organic compounds to electricity through the oxidation of these compounds into ATPs by sequential reactions [1–3]. The main difference between MFC and the other types of fuel cells is the living microorganism acts as a catalyst and oxidizes the organic materials present at the anode chamber [4–6]. Moreover, MFC is considered a clean, reliable and efficient process, and does not produce any toxic by-products [7–9]. Although the energy produced by MFC is relatively low, it possesses a unique feature in the fuel cells field; it can gain the chemical energy from several types of wastes naturally present in different environments with potential and direct conversion into electrical energy. Moreover, it can be operated at a pH close to neutrality and room temperature. Furthermore, MFC can be considered the most convenient power-generating device for wireless sensors in remote areas [10,11].

Nowadays, wireless sensors are widely used in different fields such as environmental monitoring, homeland security, oceanographic study, medical applications and military tactical surveillance in remote locations for real-time data acquisition [12]. Sometimes, recharging or replacing the batteries for the sensor is an impossible, time-consuming and/or impractical task. For resources-limited environments which have no electricity grid or limited sunlight, MFC presents a promising power source for sensors applications [8,13,14]. Moreover, the low power delivered by the MFCs is convenient for several types of sensors, as 0.5 mW is sufficient for powering the autonomous sensor [13,15].

In this regard, development of a reliable MFC to power the autonomous sensors was achieved by using sediments MFC (SMFC) [16,17]. Thomas *et al.* introduced a power management system (PMS) based on a single sediment MFC [15], Zhang *et al.* compared two laboratory-scale SMFCs with a difference in cathode arrangement (floating cathode versus bottom cathode) [18], Donovan *et al.* developed an SMFC and a PMS to run a battery-less wireless sensor [12], and Shantaram *et al.* designed an SMFC to operate electrochemical sensors and small telemetry systems to transmit the data acquired by the sensors to remote receivers [19]. Indeed, the SMFC presents a perfect power source for wireless sensors and other electronic devices, but it is still linked with the aquatic ecology and cannot be exploited in many other places such as mountains and land areas.

Rainwater contains several kinds of microorganisms collected from the atmosphere. However, based on our best knowledge, the rainwater was not previously investigated as self-containing exoelectrogens anolyte to generate power through the MFC. The rainwater microbial fuel cell (RMFC) can be used at any rainy place, so it could be a promising power generation device in the areas far from the aquatic ecology. There is ample evidence for microbiological activity in the atmosphere [20]. Kaushik *et al.* stated that the reservoir water and fresh rainwater samples exhibited a wide phylogenetic variety including genomic sequences representing Alphaproteobacteria, Actinobacteria, Betaproteobacteria, Gammaproteobacteria, Bacteriodetes and Lentisphaerae [21]. Recent studies have revealed the presence of *Escherichia coli*, *Pseudomonas aeruginosa*, *Klebsiella. pneumoniae* and *Aeromonas hydrophila* with different ratios in the examined rainwater samples [22]. These microorganisms have different roles [23], moreover, there are viable microorganisms that are capable of using formate and acetate in the atmosphere for their growth [20]. The bacteria are basic organisms in the atmosphere and play important roles in the life cycle, and they are naturally washed from the air by rain. The presence of microorganisms in the rainwater induced us to study the possibility of generating electricity through RMFC because of their highly expected ability to metabolize organic/inorganic matters. In this study, we investigate the exploitation of two different rainwater samples, based on sample collecting season, as anolytes for electricity generation through a single chamber air-cathode MFC. The results are very surprising as, although the rainwater was collected from a relatively high-quality atmosphere (Jeonju city, South Korea) [24], a promising power output could be obtained.

# 2. Material and methods

## 2.1. Rainwater samples

The samples were collected in April and December for summer and winter rainwater samples, respectively. 500 ml of each rainwater sample was collected in a sterilized glass bottle at an elevation of 59 m.a.s.l. in Jeonju, Jeollabuk-do province, Republic of Korea. 200 ml from each sample (immediately after collecting) was transferred to a 1 l sterilized bottle for microbiological (to investigate the microbial community; §3.7) and bio-electrochemical (usability as anolyte in an MFC)

analyses. These samples were transported in a cold box to the laboratory and processed within 8 h of collection. The average corresponding electrical conductivities of the collecting samples were 61 and 24 µS for the summer and winter rainwaters, respectively.

## 2.2. Anolyte preparation

The rainwater (without pretreatment) was used as anolyte without additives, and with Nutrient Broth media (1 : 1 ratio v/v) which contains (for 1 l), 15 g of peptone (Samchun Pure Chemical Co., Ltd.), 3 g of yeast extract (Samchun Pure Chemical Co., Ltd.), 1 g of glucose (Junsei Chemical Co., Ltd.) and 6 g of sodium chloride (Samchun Pure Chemical Co., Ltd.), for comparison purposes.

## 2.3. MFC construction and operation

A single chamber air-cathode MFC made of transparent polyacrylic was used as shown in electronic supplementary material, figure S1. Four centimetres between the anode and the cathode was maintained. An Ag/AgCl reference electrode is placed in the anode compartment to measure the potential of the electrodes. Cation exchange membrane (CEM, CMI-7000, Membrane International Inc., NJ, USA) was used for ions exchange. The membrane was first treated by dipping in a $H_2SO_4$ solution (0.5 M) for 18 h, and then kept in distilled water. The cathode was 2.4 cm × 2.4 cm Pt-loaded (0.5 mgcm$^{-2}$) carbon cloth (EC-20–5, Electro Chem, Inc., USA). The cathode material was in direct contact with the natural air. A carbon felt (2.4 cm × 2.4 cm, 3.18 mm, Alfa Aesar) was used as an anode. The power calculation was normalized for the anode projected surface area (6.25 cm$^2$). The current collectors were two 0.1 cm-thick high corrosion resistance stainless steel plates. The cell was assembled as described in our previous work [25] as follows: the anode was attached to an anode current collector; this assembly was placed at one side of the cell. Then, at the other side, the cathode was sandwiched between the CEM and cathode current collector. The assembled MFC was sterilized by UV (CHC LAB Co., Ltd, Korea), while the culture media and the glassware adopted in this study were autoclaved at 121°C for 20 min before operation with a steam sterilizer (Autoclave, AC-60, Hanyang Scientific Equipment Co. Ltd, Korea).

To evaluate the performance, the RMFC was driven with pure rainwater as anolyte, and with a mixture of rainwater and Nutrient broth media (1 : 1 ratio v/v). Furthermore, the aerobic and anaerobic conditions were examined and the bacteria were cultivated under anaerobic conditions by blocking the access of oxygen [26]. For the aerobic conditions, the MFC was in direct contact with the ambient air through open access of 10 mm [27]. Typically, running the RMFC under anaerobic conditions was performed by purging the anolyte using nitrogen gas bubbling for 5 min before using and closing the anolyte feeding opening in the anolyte chamber in the assembled MFC. A photo image can be found in the supporting information (electronic supplementary material, figure S1) showing the closed inlet hole. On the other hand, aerobic conditions were applied by using the anolyte without purging and opening the anolyte feeding opening; this strategy has been used according to literature [26,27]. Moreover, the temperature effect was investigated, where the RMFC was operated at the ambient temperature (0–5°C in January in Jeonju, South Korea) and at 30°C for comparison purposes.

## 2.4. Electrochemical characterization

The MFCs' electrochemical performances were evaluated using HA-151A potentiostat (HA-151A POTENTIO STAT/GALVANOSTAT, Japan). The cyclic voltammetry (CV) analysis was carried out on the assembled cell using a three-electrode set-up in which the anode was assigned as a working electrode (WE), and the cathode and Ag/AgCl as a counter (CE) and reference (RE) electrodes, respectively. The used potential window was from 1.0 to −0.4 V at a scan rate of 1 mVs$^{-1}$. The open circuit potential (OCP) values were recorded using GL220 midi-logger. After OCP stabilization, the cell circuit was closed and the power curves were obtained by linear sweep voltammetry from the maximum OCP to a zero-voltage at a scan rate of 1 mVs$^{-1}$ with a two-electrode mode, where the cathode was connected as WE and the anode as both CE and RE [28,29]. The power (P) was obtained as $P = IV$. The morphology of the used anodes was investigated by scanning electron microscopy (SEM Hitachi S-7400, Japan). The electrodes were dried at room temperature (20 ± 2°C) in sterilized Petri dishes and then used for the analysis.

## 2.5. Microbial community analyses

At the end of the experiment (after 22 days of operation), the RMFC was dismantled, and the anode was abstracted from the MFC in a sterilized laminar cabinet (CHC LAB Co., Ltd, Korea). The biofilm was sampled from both anode surface and anolyte, and then cultured in Nutrient Agar media (Sigma-Aldrich, Inc.). The biofilm was cultured to meet the DNA genomic isolation protocol requirements as the extract DNA quantity should be enough for the identification step and this required enrichment of the biofilm by culturing the biofilm [30]. The bacterial growth was subsequently isolated in a separate culture using the same media according to the growth morphology characteristic. Finally, the strains were grown overnight and processed to extract the total genomic DNA. The total genomic DNA was isolated using DNA extraction kit protocol (QIAGene, Hilden, Germany) with a little modification as described previously [30]. Briefly, 1 ml from all isolated cells were harvested in a 1.5 ml microcentrifuge tube, after centrifuging for 5 min at 8000$g$. Then, the bacterial pellet re-suspended in 180 µl buffer T1 and mixing well, followed by adding 25 µl proteinase K and vigorously homogenized by vortex and incubated at 56°C for 1–3 h with shaking during incubation. The samples after that were vortexed and a volume of 200 µl of buffer B3 was added and mixed by vigorous vortexing. A 210 µl of ethanol (100%) was added to the samples and vigorously mixed with vortex and then each sample was applied to the Nucleospin Tissue Column and placed into a collection tube and centrifuged for 1 min at 11 000$g$. After discarding the flow-through from the previous step, a 500 µl of buffer BW was added following by a centrifuge for 1 min at 11 000$g$. The Nucleospin Tissue Columns were placed in a new collection tube, 600-µl buffer B5 was added and centrifuged for 1 min at 11 000$g$. The samples were centrifuged again for 1 min at 11 000$g$. After that, the Nucleospin Tissue Columns were shifted to a 1.5 ml microcentrifuge tube and incubated for 1 min at room temperature after a 100 µl of prewarmed buffer BE (70°C) was added. Finally, the previous tubes were centrifuged for 1 min at 11 000$g$ and the flow-through was kept at −4°C for the following analysis. The DNA purity ratio was determined using 260 and 280 nm absorbance ratios. The 16S rRNA gene was amplified using forward primer 27f (5′-AGAGTTTGATCCTGGCTCAG-3′) and reverse primer 1492r (5′-GGTTACCTTGTTACGACTT-3′) by polymerase chain reaction (PCR) as previously described [31]. 16S rRNA was used for the identification of bacterial strains because it is a perfect universal target, and this gene was amplified using PCR to increase the detection sensitivity of the band when running in an electrophoresis gel. Besides that, the total bacterial count (TBC) was carried out using Nutrient agar media (Sigma-Aldrich, Inc.) [32] before as well as after utilization in the MFC for comparison purposes.

# 3. Results and discussion

## 3.1. Season and temperature effects

Basically, it is expected that the kinds of microorganisms and their quantities in the rainwater will be vary depending on the rainy season due to the change in temperature and other parameters. Therefore, the samples were collected in the summer and winter seasons. As the proposed RMFC is supposed to work within a wide temperature range through the year, so the investigation process was carried out at the ambient conditions in winter (temperature range was 0–5°C) as well as at 30°C.

Using the summer and the winter rainwaters as anolytes revealed a noteworthy difference in MFCs performance (figure 1 and electronic supplementary material, table S1). The summer rainwater MFC (SRMFC) produced a maximum OCP (588 ± 0.8 mV) with 1.5 times more than that of the winter rainwater MFC (WRMFC) at the ambient temperature after 15 days of operating time. Moreover, the OCP needs a shorter time to start up in the case of the SRMFC than that for WRMFC (figure 1a). It is noteworthy mentioning that both WRMFC and SRMFC reached almost a similar OCP after 20 days, which may be explained as depletion of the nutrients in the two media. On the other hand, at a 30°C working temperature, both of SRMFC and WRMFC, the maximum OCP (668 ± 0.9 and 763 ± 0.6 mV for the WRMFC and SRMFC, respectively) was obtained after a relatively short time (three days) compared to the ambient temperature. Furthermore, the OCP at 30°C was 1.2 and 1.7 times higher than that at the ambient temperature for SRMFC and WRMFC, respectively (electronic supplementary material, figure S2). Surprisingly, the OCP of WRMFC was superior performance compared to SRMFC only during the period of 13–20 days of operation at 30°C (figure 1b).

Overall, the OCP results indicate that the proposed RMFCs can create a very acceptable potential regardless of the collecting season and run at a relatively wide range of temperature degrees.

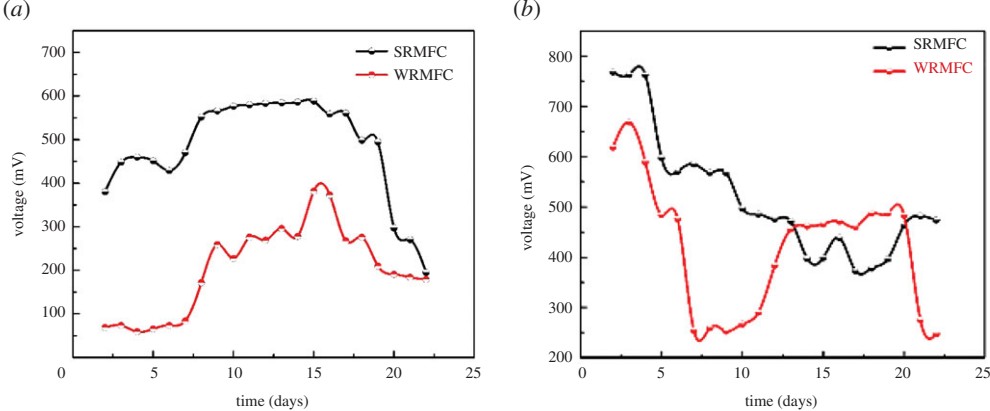

**Figure 1.** OCP for (*a*) SRMFC and WRMFC at ambient temperature; and (*b*) at 30℃.

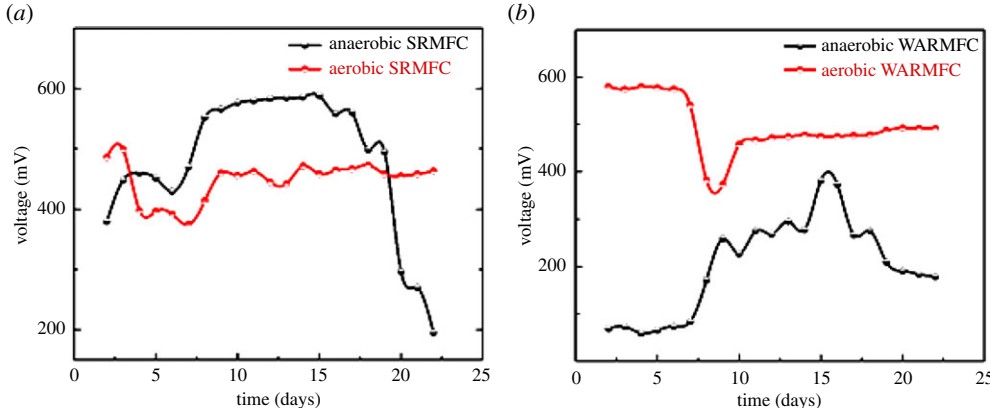

**Figure 2.** The OCP for (*a*) both anaerobic and aerobic reactors in the case of SRMFC, and (*b*) WRMFC.

Consequently, this finding suggests the possible application of the proposed RMFC in different weather conditions. However, the remarkable differences between the OCPs of SRMFC and WRMFC conclude that the summer rainwater is the favourable anolyte. The variation in the generated voltage could be attributed to the differences in the microbial community. The bacterial community (species and number) in the summer and winter rainwater has a large difference which in turn has a significant impact on RMFC performance. The bacterial community variation is clear in the bacterial identification besides the bacterial total count (see the Bacterial community analyses section).

## 3.2. Aerobic conditions effect

The proposed RMFC is suggested to work under pseudo-continuous mode in remote areas. In other words, the future design of the RMFC will be based on an open-cell configuration having the possibility of filling the cell with fresh rainwater to remove dead microorganisms and activating the biofilm as well as providing nutrients. Moreover, the maintenance of anaerobic conditions in MFC is not easy for real application [33]. Consequently, the performance of the introduced cells was investigated under aerobic conditions. Overall, MFC is an anaerobic reactor because the oxygen is acting as an inhibitor for the electroactive anaerobic microorganisms, but in the case of the facultative electroactive bacteria (which are active with both aerobic and anaerobic conditions), the MFC could be used under aerobic conditions [27]. As shown in figure 2*a*, in the case of SRMFC, the maximum OCP was $471 \pm 0.4$ mV for the aerobic conditions compared to $588 \pm 0.8$ mV for the anaerobic state. Interestingly, for the WRMFC, the aerobic reactor produces $492 \pm 0.5$ mV compared with $381 \pm 0.8$ mV for anaerobic MFC (figure 2*b*). These results indicate that the main electroactive bacteria in the SRMFC are highly active under anaerobic conditions, while in the case of the WRMFC, the main electroactive bacteria are facultative alongside the anaerobic bacteria.

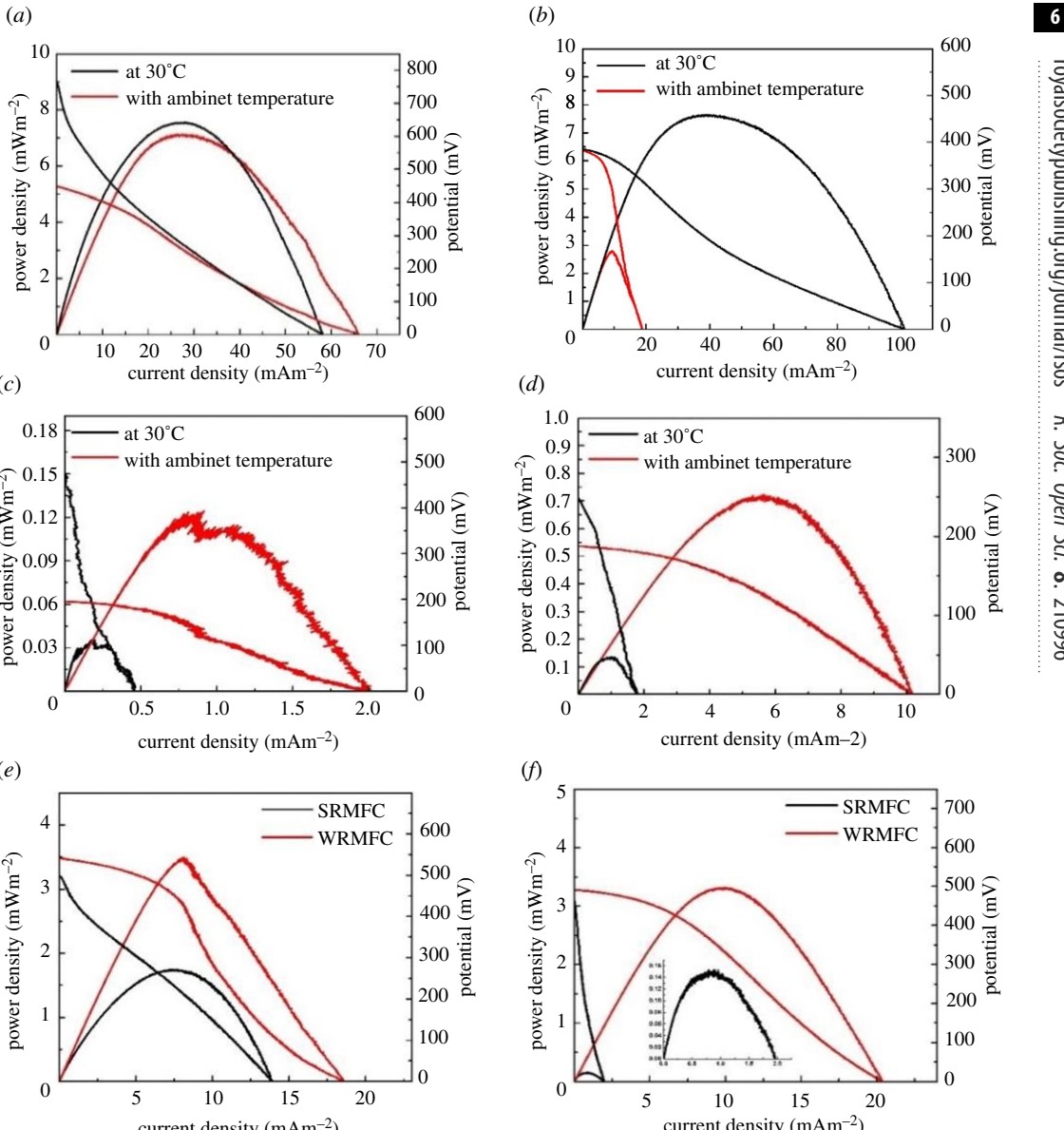

**Figure 3.** Quasi-stationary polarization and power curves for (*a*) SRMFC after three days at ambient and 30℃, under anaerobic conditions, (*b*) WRMFC after five days at 30℃ and after 15 days for ambient temperature under anaerobic conditions, (*c*) SRMFC after 22 days at ambient and 30℃ under anaerobic conditions, (*d*) WRMFC after 22 days at ambient and 30℃ under anaerobic conditions, (*e*) SRMF (after three days) and WRMFC (after seven days) at ambient temperature and under aerobic conditions, and (*f*) SRMF and WRMFC after 22 days at ambient temperature and under aerobic.

## 3.3. Power curves

The successful role of the rainwater as anolyte in MFC could be properly determined by the MFC electricity generation (figure 3). Comparatively, for SRMFC, a maximum output power of $7 \pm 0.1$ mWm$^{-2}$ at $26 \pm 0.5$ mAm$^{-2}$ and $7 \pm 0.5$ mWm$^{-2}$ at $26 \pm 0.5$ mAm$^{-2}$ was achieved under anaerobic conditions after three days at ambient temperature and 30℃, respectively (figure 3*a*). On the other hand, the WRMFC produced the maximum output power ($2 \pm 0.8$ mWm$^{-2}$ at 10 mAm$^{-2}$) after 15 days when the reactor was worked at ambient temperature, while the maximum output power ($7 \pm 0.6$ mWm$^{-2}$ at 44 mAm$^{-2}$) for 30℃ was obtained after five days, under anaerobic condition (figure 3*b*). Subsequently, in the case of SRMFC, the output power is slowly decreased to $1 \pm 0.7$ mWm$^{-2}$ (at $9 \pm 0.2$ mAm$^{-2}$) after 19 days when the cell was run with ambient temperature, while the decrease was dramatic ($1 \pm 0.1$ mWm$^{-2}$ at 5 mAm$^{-2}$) and fast (after eight days only) at 30℃ working temperature (electronic supplementary material, figure S3A). This finding can be attributed to the fast metabolization of the present nutrients at

**Table 1.** Power generation (mWm$^{-2}$) for the proposed RMFC working under anaerobic conditions and at ambient and 30℃.

| time (day) | WRMFC | | SRMFC | |
| --- | --- | --- | --- | --- |
| | ambient | 30℃ | ambient | 30℃ |
| 3 | 0.18 | 6 ± 0.41 | 7 ± 0.1 | 7 ± 0.5 |
| 5 | 0.15 | 7 ± 0.6 | 6 ± 0.2 | 2 ± 0.67 |
| 8 | 0.59 | 0.84 | 6 | 1 ± 0.1 |
| 15 | 2 ± 0.8 | 0.52 | 3 | 0.059 |
| 19 | 0.53 ± 0.03 | 0.48 ± 0.02 | 1 ± 0.7 | 0.04 |
| 22 | 0.72 ± 0.01 | 0.13 | 0.12 ± 0.01 | 0.035 |

30℃. While in the case of WRMFC, the power output was strongly affected after 19 days achieving 0.53 ± 0.03 mWm$^{-2}$ at 4 ± 0.1 mAm$^{-2}$ and 0.48 ± 0.02 mWm$^{-2}$ at 2 ± 0.1 mAm$^{-2}$ for the ambient and 30℃ temperature, respectively, (electronic supplementary material, figure S3B).

Finally, after 22 days, the output power was limited to 0.12 ± 0.01 mWm$^{-2}$ at 0.85 ± 0.02 mAm$^{-2}$ and 0.035 mWm$^{-2}$ at 0.22 ± 0.02 mAm$^{-2}$ for SRMFC at ambient and 30℃ temperatures, respectively, (figure 3c). Whereas in the case of the WRMFC, the output power was 0.72 ± 0.01 mWm$^{-2}$ at 6 ± 0.1 mAm$^{-2}$ and 0.13 mWm$^{-2}$ at 1 mAm$^{-2}$ at ambient and 30℃ temperatures, respectively (figure 3d).

Interestingly, under aerobic conditions and at ambient temperature (figure 3e,f), the maximum power output for the WRMFC was achieved after seven days (3 ± 0.49 mWm$^{-2}$ at 7 ± 0.2 mAm$^{-2}$) which in turn was two times more than the maximum output power (1 ± 0.74 mWm$^{-2}$ at 7 ± 0.2 mAm$^{-2}$) for SRMFC after three days (figure 3e). Furthermore, after 22 days, the SRMFC output power was limited to 0.15 mWm$^{-2}$ at 1 mAm$^{-2}$, while the WRMFC output power is sustained close to the maximum power (3 ± 0.3 mWm$^{-2}$ at 10 ± 0.3 mAm$^{-2}$; figure 3f).

The output power values emphasize that the RMFC can be used with either summer or winter rainwater as anolyte and under both aerobic and anaerobic conditions. Moreover, these results show that the season of the anolyte collection, as well as the operation conditions (temperature degree and aerobicity), have a distinct influence on the MFC output power. The summer rainwater is the favourable anolyte with the ambient temperature where it achieved output power higher than that for the winter rainwater with 2.5 times (around 7 and 3 mWm$^{-2}$, respectively). On the contrary, the winter rainwater is the proper anolyte in the case of the aerobic condition, where the WRMFC power output was 22-fold higher than the power output for the SRMFC (around 3 and 0.15 mWm$^{-2}$, respectively). Tables 1 and 2 summarize the obtained powers from the proposed RMFCs.

Although, it seems that the generated power from the proposed RMFCs is small, however using the rainwater without inoculation strongly recommends exploiting these cells as power sources in the remote areas as duplicating the power can be achieved by assembling the individual cells in a stack. Moreover, the successful running of the proposed microbial fuel cells under aerobic conditions changes the proposed cells to be a sustainable source of power. Simply, if the constructed cells were left open to the atmosphere, the cells will be loaded by fresh rainwater which changes the system to be a continuous power source.

## 3.4. Reliability evaluation

The addition of nutrients to enrich the rainwater flora is possible in laboratory experiments, but it is a difficult task in real applications. Therefore, the summer rainwater (alone without additives) was invoked to evaluate the feasibility of using a pristine rainwater as anolyte in the proposed MFC at an ambient temperature in the winter (0–5℃).

Interestingly, the rainwater with and without nutrients showed almost the same progression with close maximum voltage output (figure 4a). The summer rainwater alone (SRMFC-A, for short) produced a maximum voltage output of 553 ± 0.6 mV, with only a 50 mV decrease compared to the maximum output of the summer rainwater with nutrients (588 mV). Furthermore, the OCP progression rate of summer rainwater with nutrients (SRMFC-WN, for short) needs a short time to start up and achieve the maximum output OCP (six and 16 days, respectively), while SRMFC-A required more than 14 days to start up, where the maximum OCP was achieved at 18 days.

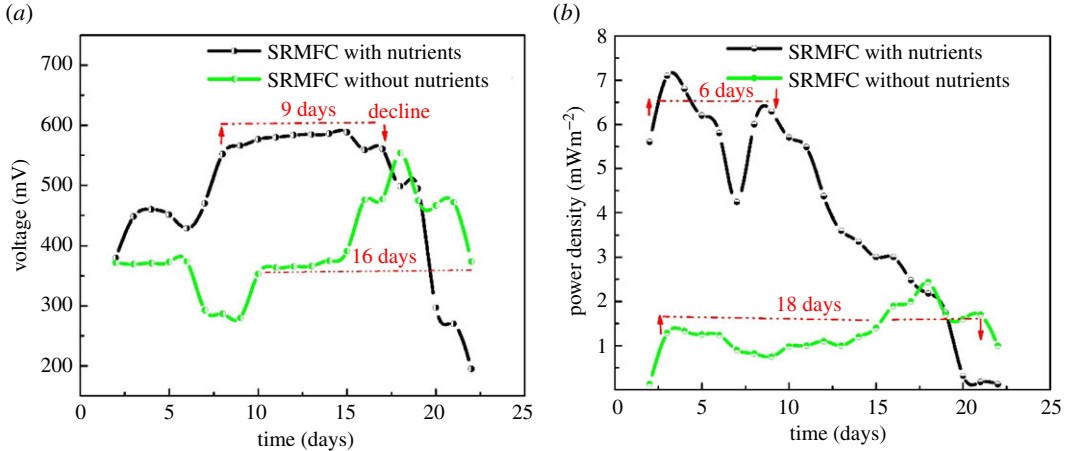

**Figure 4.** The SRMFC with and without nutrients, (*a*) voltage output, and (*b*) power density, throughout the operation period.

**Table 2.** Influence of aerobic condition on the generated (mWm$^{-2}$) for the proposed RMFC at ambient temperature.

| time (day) | WRMFC | SRMFC |
|---|---|---|
| 3 | $2 \pm 0.2$ | $1 \pm 0.74$ |
| 7 | $3 \pm 0.49$ | 0.34 |
| 22 | $3 \pm 0.3$ | 0.15 |

In the case of the SRMFC-A, the maximum power output ($2.44 \pm 0.05$ mWm$^{-2}$ at $10 \pm 0.5$ mAm$^{-2}$) was observed after 18 days, while the SRMFC-WN achieved the maximum power output ($7 \pm 0.1$ mWm$^{-2}$ at $26 \pm 0.5$ mAm$^{-2}$) after three days (electronic supplementary material, figure S4B). The short startup of RMFC-WN could be ascribed to the effect of nutrients regarding the bacterial growth curve [34], where the enrichment of rainwater accelerated the microbial flora growth rate to finish the lag phase and start the log and then stationary phases in a short time. This acceleration influence appeared to be a passive effect for sustaining SRMFC-WN, whereas the stationary phase starts rapidly, it has been also terminated rapidly and the OCP and power output declined steadily after achieving their maximum outputs directly (at 16 and nine days of operation, respectively). Therefore, the SRMFC-WN sustained the voltage output at the same rate for nine days and just six days for power generation through the operation period (figure 4*b* and electronic supplementary material, figure S4A). Overall, these results revealed that the SRMFC alone without nutrients could produce a reasonable voltage output with a wide range of temperature degrees, and the possibility of the proposed RMFC in a real application accordingly.

## 3.5. Cyclic voltammetry

The CV analysis can be exploited as a powerful methodology to check the dependency of the electron transfer on mediators [35]. Briefly, the mediators (also known as shuttles) are considered shuttle carriers for the electrons from the microorganism surface to the anode. The electron transfer takes place in the form of redox reactions. In other words, the mediators are reduced at the microorganism surface and oxidized at the anode. Therefore, the presence of redox peaks in the CV cycle can be assigned to the presence of mediators [36,37]. Cyclic voltammograms of the RMFC were carried out at 1 mVm$^{-1}$ scan rate. As shown in figure 5, smooth and peaks-free cycles were obtained for both SRMFC and WRMFC. Accordingly, it can be claimed that the electrons are delivered from the bacterial biofilm to the anode surface by direct contact and without mediators [2].

## 3.6. Scanning electron microscopy (SEM) analysis

SEM analysis (figure 6) was carried out at the end of the experiments to check the bacterial biofilm morphology on the anode surface. In general, a thick and good-structured biofilm could be observed

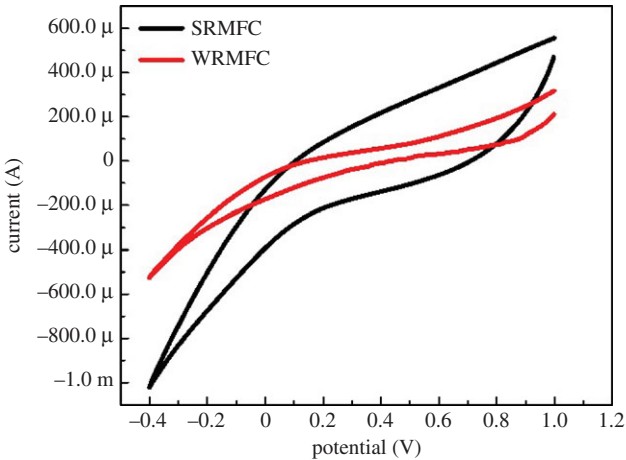

**Figure 5.** CV curves for the SRMFC and WRMFC at 1 mVs$^{-1}$ scan rate after OCP stabilization.

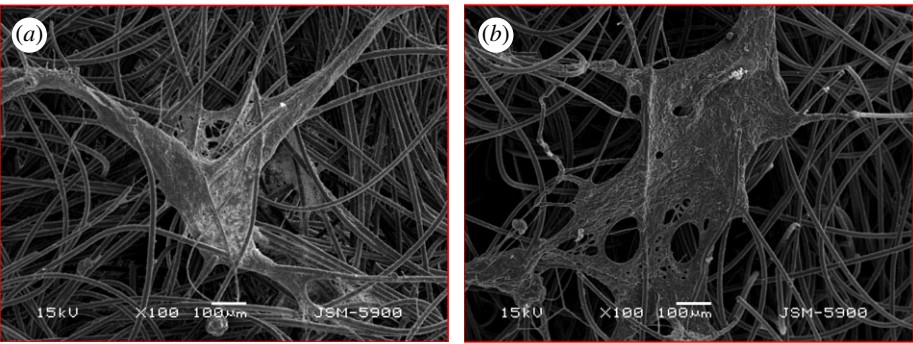

**Figure 6.** SEM images for bacterial biofilm for both; (*a*) SRMFC and (*b*) and WRMFC.

for both SRMFC and WRMFC, and there were no fundamental differences between their biofilm's appearance. These images revealed that the rainwater flora contains electroactive bacteria that successfully attached closely to the carbon felt's thread surface and developed an electroactive biofilm which in turn transmits electrons to the anode.

Biofilm uniformity at the anode surface indicates a non-fermentable fuel feeding [38], while fermentable fuel feeding is associated with microbial clumps appearance in the anodic biofilm [39]. The uniformity of observed biofilm of both SRMFC and WRMFC represent a non-fermentative electroactive bacteria, where rainwater contains simple compounds (formate, acetate, etc.) which fermentation needs before microbial community consumption [40].

## 3.7. Bacterial community analyses

### 3.7.1. Bacterial population

Before using in the RMFCs, the as-collected rainwater samples have been examined using microbiological analyses to investigate the microbial community in the used rainwater. The TBC is showing different bacterial loads as presented in table 3. The variation in the TBC is related to the sampling season (summer and winter). TBC clarified that the bacterial load in the winter rainwater sample is less than that in the summer one which in turn could be an acceptable reason alongside the bacterial species for the variation in the power output between the two samples.

### 3.7.2. Bacterial diversity of RMFC

Synergetic interactions between microorganisms present an important role in the formation of electroactive biofilms, which in turn, is the main key in the longevity and success of bio-electrochemical systems [41]. The sequences with the similarity ratio to strains in both SRMF and

**Table 3.** The TBC for the summer and winter rainwater samples before and after using with MFC.

| before using | | after using | |
|---|---|---|---|
| summer rainwater | winter rainwater | SRMFC | WRMFC |
| $5.14 \times 10^3$ | $4.19 \times 10^3$ | $8.12 \times 10^3$ | $7.13 \times 10^3$ |

**Table 4.** Identification of 11 DGGE bands based on 16S rRNA genes, from the anode surface biofilm and anolyte.

| band | closest sequence | similarity (%) | accession no. | cited isolation source |
|---|---|---|---|---|
| 1 | Lactobacillus coryniformis strain 17-9 | 99 | NR_029018.1 | summer rainwater |
| 2 | Lactobacillus coryniformis strain DPP-LP3 | 93 | NR_029018.1 | summer rainwater |
| 3 | Lactobacillus coryniformis strain KLDS 1.0723 | 99 | NR_029018.1 | summer rainwater |
| 4 | Bacillus cereus strain P2 | 99 | NR_074540.1 | summer rainwater |
| 5 | Bacillus methylotrophicus strain CBMB205 | 99 | NR_116240.1 | summer rainwater |
| 6 | Lactobacillus coryniformis strain 17-9 | 99 | NR_029018.1 | summer rainwater |
| 7 | Staphylococcus capitis strain Y85XJ04 | 95 | NR_113348.1 | summer rainwater |
| 8 | Staphylococcus capitis strain ATCC 27840 | 99 | NR_117006.1 | summer rainwater |
| 9 | Staphylococcus capitis-Y41 | 99 | JX_094948.1 | winter rainwater |
| 10 | Staphylococcus capitis strain BQEN3-03 | 98 | FJ_380955.1 | winter rainwater |
| 11 | Enterobacteriaceae bacterium strain 4 | 97 | KY_681857.1 | winter rainwater |
| 12 | Staphylococcus capitis strain 309-16 | 98 | MG_557816.1 | winter rainwater |
| 13 | Staphylococcus capitis strain BBN3T-04d | 98 | FJ_357614.1 | winter rainwater |
| 14 | Pseudomonas sp. CSJ-3 | 98 | KF_861966 | winter rainwater |

WRMFC is presented in table 4. It is clear that *Lactobacillus* sp. (electronic supplementary material, table S2) is the main electroactive bacteria in SRMFC, which was previously stated as electroactive bacteria in the MFC application [42]. Besides, *Bacillus* sp. and *Staphylococcus* sp. are present in the SRMFC bacterial community, and their role as the electroactive effect was also previously reported [41,43]. However, *Staphylococcus* sp. (electronic supplementary material, table S3) is the dominant genus and main electroactive in the case of the WRMFC. Furthermore, *Pseudomonas* spp., which is well known as an electroactive genus in MFC[45], was detected in the WRMFC. Moreover, the *Enterobacteriaceae* bacterium strain was also found in the investigated sample, however, its role as electroactive bacteria still needs more study.

Overall, the proposed MFC is applicable in many countries due to the high average raining days around the year based on the world weather and climate website [44]. Electronic supplementary material, figure S5 shows the average daily water in some randomly selected countries, however, on the website, the average of rainy days could be found for many countries. Practically, the proposed RWMFC can work in the remote area by first filling the cell with the rainwater and leaving it at open circuit mode until the formation of the biofilm at the anode surface and stability of OCV. Later, any additional rainwater that fills the cell will be used as a source of nutrients for the microorganisms to generate electricity.

## 4. Conclusion

Rainwater can be effectively exploited as an anolyte in an air-cathode single-chamber MFC. Based on the results, it is preferable to use collected water within the summer season for the anaerobic conditions MFC. On the other hand, winter rainwater is recommended for aerobic reactors. In the case of the summer rainwater reactor, the dominant electroactive bacteria were *Lactobacillus* spp. However, the main electroactive genus in the winter rainwater reactor was *Staphylococcus* spp. The investigated

rainwater-driven microbial fuel cells might be used not only for powering wireless sensors but also as a biosensing tool to monitor the atmosphere microbial community and the ratio of different elements and their toxicity [36]. Based on our best knowledge, the rainwater was not previously investigated as anolyte for the MFC; consequently, this area needs more investigation to adopt this valuable source of energy and overcome the obstacles.

Data accessibility. Data are available from the Dryad Digital Repository: http://doi.org/10.5061/dryad.pvmcvdnm7 [45]. The data are provided in the electronic supplementary material [46].

Authors' contributions. M.T.A. carried out the laboratory work, participated in data analysis, carried out sequence alignments, participated in the design of the study and drafted the manuscript; A.S.Y. participated in the statistical analyses; M.I.H. and M.A.H.M.J. participated in the biological study; S.-T.H. participated in the biological study and discussed their results; N.A.M.B. conceived of the study, designed the study, coordinated the study and helped draft the manuscript.

Competing interests. We declare we have no competing interests.

Funding. We received no funding for this study.

Acknowledgements. The authors wish to thank the Royal Society Open Science Editorial Office, associate editors, and the anonymous reviewers for the quality of their feedback and very insightful comments that have contributed to the final version of the paper.

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
