## [Peer Review File · Royal Society Open Science]

Review History

RSOS-210996.R0 (Original submission)

Review form: Reviewer 1

Is the manuscript scientifically sound in its present form?

Yes

Are the interpretations and conclusions justified by the results?

Yes

Is the language acceptable?

Yes

Do you have any ethical concerns with this paper?

No

Have you any concerns about statistical analyses in this paper?

No

Recommendation?

Major revision is needed (please make suggestions in comments)

Comments to the Author(s)

It's interesting that Amen et al. carried out the investigation on the rainwater as the anolyte in an air-cathode microbial fuel cell (MFC) to generate electricity. They found the season and the oxygen might be the important factors that affect the performance of rainwater-driven MFC. This try is very useful for the power supply of some sensors in mountains and land areas. However, I still have some confusions about the design and the explanations. Why the authors did not inoculate the MFC and start up? No more microbial and chemical information on the rainwater? The authors should do more work on the microbial names and formats. Overall, it can't be accepted for publication before a major revision.

Here are my specific comments.

Abstract

P2, line5, 'aerobic or anaerobic conditions' is not clear;

P2, line 14-15, Authors should confirm all microbial names and their formats, almost all names are not correct, such as, 'lactobacillus sp.', 'Staphylococcus sp. is the main electroactive bacteria';

P2, line 15-17, 'cyclic voltammetry analysis confirms that the electrons are delivered directly from the bacterial biofilm to the anode surface and without mediators.' Your data support this conclusion?

Introduction

P4, line8, 'It is known that rainwater contains several kinds of microorganisms collected from the atmosphere.' Who knows it?

P4, line18, 'some bacteria can catalyze ice formation at temperatures near 19 -2 °C'? confused about this description;

P5, line4-5, wrong formats;

Materials and Methods

P5, line12-13, here the authors addressed rainwater 'was transferred to 1 L sterilised bottle for chemical and microbiological analyses.', however, we did not find the chemical and microbiological results;

Section '2.3. MFC Construction and operation' MFC need inoculation and a long time to startup, however, you did not address the methods, no results.

P6, line19-23, The nutrient in the control is too high 'with mixture of rainwater and Nutrient broth media (1:1 ratio v/v)'; the aerobic and anaerobic conditions are not clear;

P7, line18-22, experiment design is not acceptable. The authors should analyze the biofilm directly, but not cultured the biofilm;

P8, line17-19, what the purpose for amplifying the 16S rRNA gene?

Results and Discussion

P9, line10 and others, 'the ambient temperature' should give a temperature arrange;

P10, section '3.2. Aerobic Conditions Effect', how to maintain the anaerobic conditions? What the difference with the aerobic condition?

P19, section' 3.7.2. Bacterial Diversity of RMFC', confirm all the formats of the microbial names, such as the lactobacillus sp., staphylococcus sp., Staphylococcus sp.....need more work to do;

Table 2, Table S2 and S3 also should be corrected in formats.

Review form: Reviewer 2

Is the manuscript scientifically sound in its present form?

Yes

Are the interpretations and conclusions justified by the results?

Yes

Is the language acceptable?

Yes

Do you have any ethical concerns with this paper?

No

Have you any concerns about statistical analyses in this paper?

No

Recommendation?

Accept with minor revision (please list in comments)

Comments to the Author(s)

I think that the idea presented in the paper is very original and worth keep exploring in the future.

The only aspect I think should be considered in the paper is an estimation about the energy produced by the Fuel cells. Authors are focused in monitoring voltage produced in the fuel cell, but as the intended use of such a device is to provide power, voltage is not sufficient. The authors measure an I-V curve every now and then, but this is not sufficient as the power delivered by a system with no forced convection will not be constant. Also the lazy metabolism of bacteria may prevent the system from producing the current reported in the I-V in a continuous manner.

Therefore I think that a discussion and some estimations about energy generated in the cells in such a long period of time is mandatory. I agree that sensors can operated with only few mW but for how long? How big should a FC be to continuously provide power to such sensors?

Are there any pictures of the fuel cells used in the experiment? Can the authors also describe how an operative system in the field would operate? Would it collect only some rain water at the beginning of the season or would the system accept rain through the whole period?

Decision letter (RSOS-210996.R0)

Dear Dr Amen

The Editors assigned to your paper RSOS-210996 "Rainwater-driven Microbial Fuel Cells for Power Generation in the Remote Areas" have now received comments from reviewers and would like you to revise the paper in accordance with the reviewer comments and any comments from the Editors. Please note this decision does not guarantee eventual acceptance.

Please submit your revised manuscript and required files (see below) no later than 21 days from today's (ie 07-Sep-2021) date. Note: the ScholarOne system will 'lock' if submission of the revision is attempted 21 or more days after the deadline. If you do not think you will be able to meet this deadline please contact the editorial office immediately.

on behalf of Prof Pete Smith (Subject Editor)
openscience@royalsociety.org

Associate Editor Comments to Author:

Two reviewers have provided commentary on your work - please ensure that your revision carefully addresses their comments and, where additional clarifications or content is requested, you include it in the manuscript resubmission, too. Please note that the reviewers will be invited to assess your revision.

Reviewer comments to Author:

Reviewer: 1

Comments to the Author(s)

It's interesting that Amen et al. carried out the investigation on the rainwater as the anolyte in an air-cathode microbial fuel cell (MFC) to generate electricity. They found the season and the oxygen might be the important factors that affect the performance of rainwater-driven MFC. This try is very useful for the power supply of some sensors in mountains and land areas. However, I still have some confusions about the design and the explanations. Why the authors did not inoculate the MFC and start up? No more microbial and chemical information on the rainwater? The authors should do more work on the microbial names and formats. Overall, it can't be accepted for publication before a major revision.

Here are my specific comments.

Abstract

P2, line5, 'aerobic or anaerobic conditions' is not clear;

P2, line 14-15, Authors should confirm all microbial names and their formats, almost all names are not correct, such as, 'lactobacillus sp.', 'Staphylococcus sp. is the main electroactive bacteria';

P2, line 15-17, 'cyclic voltammetry analysis confirms that the electrons are delivered directly from the bacterial biofilm to the anode surface and without mediators.' Your data support this conclusion?

Introduction

P4, line8, 'It is known that rainwater contains several kinds of microorganisms collected from the atmosphere.' Who knows it?

P4, line18, 'some bacteria can catalyze ice formation at temperatures near 19 -2 °C'? confused about this description;

P5, line4-5, wrong formats;

Materials and Methods

P5, line12-13, here the authors addressed rainwater 'was transferred to 1 L sterilised bottle for chemical and microbiological analyses.', however, we did not find the chemical and microbiological results;

Section '2.3. MFC Construction and operation' MFC need inoculation and a long time to startup, however, you did not address the methods, no results.

P6, line19-23, The nutrient in the control is too high 'with mixture of rainwater and Nutrient broth media (1:1 ratio v/v)'; the aerobic and anaerobic conditions are not clear;

P7, line18-22, experiment design is not acceptable. The authors should analyze the biofilm directly, but not cultured the biofilm;

P8, line17-19, what the purpose for amplifying the 16S rRNA gene?

Results and Discussion

P9, line10 and others, 'the ambient temperature' should give a temperature arrange;

P10, section '3.2. Aerobic Conditions Effect', how to maintain the anaerobic conditions? What the difference with the aerobic condition?

P19, section' 3.7.2. Bacterial Diversity of RMFC', confirm all the formats of the microbial names, such as the lactobacillus sp., staphylococcus sp., Staphylococcus sp.....need more work to do;

Table 2, Table S2 and S3 also should be corrected in formats.

Reviewer: 2

Comments to the Author(s)

I think that the idea presented in the paper is very original and worth keep exploring in the future.

The only aspect I think should be considered in the paper is an estimation about the energy produced by the Fuel cells. Authors are focused in monitoring voltage produced in the fuel cell, but as the intended use of such a device is to provide power, voltage is not sufficient. The authors measure an I-V curve every now and then, but this is not sufficient as the power delivered by a system with no forced convection will not be constant. Also the lazy metabolism of bacteria may prevent the system from producing the current reported in the I-V in a continuous manner.

Therefore I think that a discussion and some estimations about energy generated in the cells in such a long period of time is mandatory. I agree that sensors can operated with only few mW but for how long? How big should a FC be to continuously provide power to such sensors?

Are there any pictures of the fuel cells used in the experiment? Can the authors also describe how an operative system in the field would operate? Would it collect only some rain water at the beginning of the season or would the system accept rain through the whole period?

===PREPARING YOUR MANUSCRIPT===

one version identifying all the changes that have been made (for instance, in coloured highlight, in bold text, or tracked changes);
 a 'clean' version of the new manuscript that incorporates the changes made, but does not highlight them. This version will be used for typesetting if your manuscript is accepted.

===PREPARING YOUR REVISION IN SCHOLARONE===

- Any electronic supplementary material (ESM).
- If you are requesting a discretionary waiver for the article processing charge, the waiver form must be included at this step.
- If you are providing image files for potential cover images, please upload these at this step, and inform the editorial office you have done so. You must hold the copyright to any image provided.
- A copy of your point-by-point response to referees and Editors. This will expedite the preparation of your proof.

- Ensure that your data access statement meets the requirements at <https://royalsociety.org/journals/authors/author-guidelines/#data>. You should ensure that you cite the dataset in your reference list. If you have deposited data etc in the Dryad repository, please include both the 'For publication' link and 'For review' link at this stage.
- If you are requesting an article processing charge waiver, you must select the relevant waiver option (if requesting a discretionary waiver, the form should have been uploaded at Step 3 'File upload' above).
- If you have uploaded ESM files, please ensure you follow the guidance at <https://royalsociety.org/journals/authors/author-guidelines/#supplementary-material> to include a suitable title and informative caption. An example of appropriate titling and captioning may be found at https://figshare.com/articles/Table_S2_from_Is_there_a_trade-off_between_peak_performance_and_performance_breadth_across_temperatures_for_aerobic_scope_in_teleost_fishes_/3843624.

Author's Response to Decision Letter for (RSOS-210996.R0)

See Appendix A.

RSOS-210996.R1 (Revision)

Review form: Reviewer 1

Is the manuscript scientifically sound in its present form?

Yes

Are the interpretations and conclusions justified by the results?

Yes

Is the language acceptable?

Yes

Do you have any ethical concerns with this paper?

No

Have you any concerns about statistical analyses in this paper?

No

Recommendation?

Accept with minor revision (please list in comments)

Comments to the Author(s)

I received the response of the authors on my concerns, almost all questions have been addressed and resolved. One more thing, you should know that the microbial community structure of the cultured biofilm is totally different from the real structure of anodic biofilm. You should pay more attention on the formats for the microbial names, such as the strains in 'Table 4 ' should not be in the italic, s should be capital in '16s'.

Review form: Reviewer 2

Is the manuscript scientifically sound in its present form?

Yes

Are the interpretations and conclusions justified by the results?

Yes

Is the language acceptable?

Yes

Do you have any ethical concerns with this paper?

No

Have you any concerns about statistical analyses in this paper?

No

Recommendation?

Accept as is

Comments to the Author(s)

The paper can now be published

Decision letter (RSOS-210996.R1)

Dear Dr Amen

On behalf of the Editors, we are pleased to inform you that your Manuscript RSOS-210996.R1 "Rainwater-driven Microbial Fuel Cells for Power Generation in the Remote Areas" has been accepted for publication in Royal Society Open Science subject to minor revision in accordance

with the referees' reports. Please find the referees' comments along with any feedback from the Editors below my signature.

Please submit your revised manuscript and required files (see below) no later than 7 days from today's (ie 22-Oct-2021) date. Note: the ScholarOne system will 'lock' if submission of the revision is attempted 7 or more days after the deadline. If you do not think you will be able to meet this deadline please contact the editorial office immediately.

on behalf of Pete Smith (Subject Editor)
openscience@royalsociety.org

Reviewer comments to Author:

Reviewer: 1

Comments to the Author(s)

I received the response of the authors on my concerns, almost all questions have been addressed and resolved. One more thing, you should know that the microbial community structure of the cultured biofilm is totally different from the real structure of anodic biofilm. You should pay more attention on the formats for the microbial names, such as the strains in 'Table 4' should not be in the italic, s should be capital in '16s'.

Reviewer: 2

Comments to the Author(s)

The paper can now be published

===PREPARING YOUR MANUSCRIPT===

one version should clearly identify all the changes that have been made (for instance, in coloured highlight, in bold text, or tracked changes);
 a 'clean' version of the new manuscript that incorporates the changes made, but does not highlight them. This version will be used for typesetting.

===PREPARING YOUR REVISION IN SCHOLARONE===

- An editable file of all figure and table captions.
- Note: you may upload the figure, table, and caption files in a single Zip folder.
- Any electronic supplementary material (ESM).
 - If you are requesting a discretionary waiver for the article processing charge, the waiver form must be included at this step.
 - If you are providing image files for potential cover images, please upload these at this step, and inform the editorial office you have done so. You must hold the copyright to any image provided.
 - A copy of your point-by-point response to referees and Editors. This will expedite the preparation of your proof.

- Ensure that your data access statement meets the requirements at <https://royalsociety.org/journals/authors/author-guidelines/#data>. You should ensure that you cite the dataset in your reference list. If you have deposited data etc in the Dryad repository, please only include the 'For publication' link at this stage. You should remove the 'For review' link.
- If you are requesting an article processing charge waiver, you must select the relevant waiver option (if requesting a discretionary waiver, the form should have been uploaded, see 'File upload' above).
- If you have uploaded any electronic supplementary (ESM) files, please ensure you follow the guidance at <https://royalsociety.org/journals/authors/author-guidelines/#supplementary-material> to include a suitable title and informative caption. An example of appropriate titling and captioning may be found at https://figshare.com/articles/Table_S2_from_Is_there_a_trade-off_between_peak_performance_and_performance_breadth_across_temperatures_for_aerobic_scope_in_teleost_fishes_/3843624.

Author's Response to Decision Letter for (RSOS-210996.R1)

See Appendix B.

Decision letter (RSOS-210996.R2)

Dear Dr Amen,

I am pleased to inform you that your manuscript entitled "Rainwater-driven Microbial Fuel Cells for Power Generation in the Remote Areas" is now accepted for publication in Royal Society Open Science.

on behalf of Professor Pete Smith (Subject Editor)
openscience@royalsociety.org

Appendix A

Dear Prof. Pete Smith

Subject Editor of Royal Society Open Science Journal

Thank you for your kind response about the manuscript (ID **RSOS-210996**) titled "**Rainwater-driven Microbial Fuel Cells for Power Generation in the Remote Areas**".

The referees' comments supported the mentioned novelty in the cover letter. Accordingly, we can confidently claim that this manuscript will have strong impact in the microbial fuel cells field as it introduces a new class of microbial fuel cells based on the rainwater.

Actually, the given comments were also helpful to strength the manuscript. We would like to inform you that we have modified the manuscript according to the given comments.

To make it more easily, we have written the comments in bold phase followed by the responses in normal one. Moreover, in the revised manuscript, you can find the changes in the text in blue color.

We hope our responses cover all the comments. It will be our pleasure to respond about any more comments.

Thank you for your cooperation

Sincerely yours

Nasser A. M. Barakat

Professor

Minia University

Reviewer: 1

Comments to the Author(s)

It's interesting that Amen et al. carried out the investigation on the rainwater as the anolyte in an air-cathode microbial fuel cell (MFC) to generate electricity. They found the season and the oxygen might be the important factors that affect the performance of rainwater-driven MFC. This try is very useful for the power supply of some sensors in mountains and land areas. However, I still have some confusions about the design and the explanations. Why the authors did not inoculate the MFC and start up? No more microbial and chemical information on the rainwater? The authors should do more work on the microbial names and formats. Overall, it can't be accepted for publication before a major revision.

Here are my specific comments.

First, we strongly appreciate the very valuable comments given by the reviewer. We believe that the given comments were generated because of problems in the explanation of some parts in the original manuscript. Below, we introduce a detailed response for every point.

Abstract

1. P2, line5, 'aerobic or anaerobic conditions is not clear.

Response: We agree with the reviewer, the original text was not clear. The text has been updated and highlighted in the revised manuscript; *page 2 line 5.*

2. P2, line 14-15, Authors should confirm all microbial names and their formats, almost all names are not correct, such as, 'lactobacillus sp.', 'Staphylococcus sp. is the main electroactive bacteria.

Response: All microbial names have been checked and corrected in the whole manuscript.

3. P2, line 15-17, 'cyclic voltammetry analysis confirms that the electrons are delivered

directly from the bacterial biofilm to the anode surface and without mediators.’ Your data support this conclusion?

Response: According to literature, the cyclic voltammetry analysis can be exploited to check the dependency of the electron transfer on mediators; for instance, Zou et. al, (ref. 35) and references therein, stated this hypothesis. Grattieri et. al, (ref. 36) scientifically explained this conclusion as the mediators can transfer the electrons from the cell membrane to the anode surface through a redox reaction(s). Accordingly, appearance of redox peak(s) in the cyclic voltammograms indicates presence of mediator(s) otherwise the electron transfer in the investigated MFC is a mediator-less process (it is also approved by B.E Logan (ref. 2)). As shown in Fig. 5, smooth and peaks-free cycles for both SRMFC and WRMFC, so we claimed that the electrons are delivered from the bacterial biofilm to the anode surface by direct contact and without mediators.

To avoid readers' confusion, section 3.5 in the revised manuscript has been updated.

Introduction

P4, line8, ‘It is known that rainwater contains several kinds of microorganisms collected from the atmosphere.’ Who knows it?

Response: It is known that microorganisms can be found in any medium including the air atmosphere. Therefore, it is accepted to claim that the rainwater contains several kinds of microorganisms coming from the air. In the revised manuscript, this hypothesis was paraphrased and supported by some references; 20 to 23.

P4, line18, ‘some bacteria can catalyse ice formation at temperatures near 19 –2 °C’? confused about this description.

Response: Actually, this statement has been cited from ref. 23 (B. C. Christner et al, Science, 2008, 319, 1214). However, to avoid any confusion, the statement has been deleted in the revised manuscript especially it does not affect the manuscript main topic.

P5, line4-5, wrong formats.

Response: The text has been updated.

Materials and Methods

P5, line12-13, here the authors addressed rainwater ‘was transferred to 1 L sterilised bottle for chemical and microbiological analyses.’, however, we did not find the chemical and microbiological results.

Response: According to this valuable comment, section 3.7, which explains the microbiological analysis and their result (bacterial community analyses), has been updated.

Before using in the RMFCs, the as-collected rainwater samples have been examined using microbiological analyses to investigate the microbial community in the used rainwater.

On the other hand, chemical analysis was a wrong terminology; it should be "bioelectrochemical" as this water was investigated for power generation in the proposed RMFC.

The text has been updated; “chemical and microbiological analyses” has been changed to “**microbiological (to investigate the microbial community; section 3.7) and bioelectrochemical (usability as anolyte in an MFC) analyses.**”

Section ‘2.3. MFC Construction and operation’ MFC need inoculation and a long time to start up, however, you did not address the methods, no results.

Response: As the reviewer realized in his general comment, the introduced manuscript investigates utilization of rainwater as anolyte for the MFC to be exploited as a small power source in the remote areas. In these remote areas, it is difficult to inoculate the MFC or use fed-batch mode. Therefore, the experimental work in this study was planned on using the naturally present microbial community in the rainwater as bio-catalysts in the proposed RMFC without any external inoculation. Moreover, a batch mode was selected to operate the proposed MFC. Overall, the manuscript is opening an avenue for the researchers to establish a new class of MFCs based on the rainwaters.

P6, line19-23, The nutrient in the control is too high ‘with mixture of rainwater and Nutrient broth media (1:1 ratio v/v)’

Response: This ratio has been used to compare the maximum effect of nutrients (which should be high) with rainwater alone as has been discussed in section 3.4 titled “reliability and evaluation”. It is mentioned in this section that “the addition of nutrients to enrich the rainwater flora is possible in the lab experiments, but it’s a difficult task in the real applications. Therefore, the summer rainwater (alone without additives) was invoked to evaluate the feasibility of using pristine rainwater as an anolyte in the proposed MFC.

The aerobic and anaerobic conditions are not clear.

Response: We agree with the reviewer, this part was not clear in the original manuscript.

Typically, running the RMFC under anaerobic conditions was performed by purging first the anolyte using nitrogen gas bubbling for 5 min before using in the RMFC and closing the anolyte feeding hole in the anolyte chamber in the assembled MFC. A photo image can be found in the supporting information (Fig. S1) showing the closed inlet opening. On the other hand, aerobic conditions were applied by utilizing the anolyte without purging and opening the anolyte feeding opening; this strategy has been used according to literature (refs. 26 & 27).

This explanation has been added in the revised manuscript, page 7, line 5.

P7, line18-22, experiment design is not acceptable. The authors should analyse the biofilm directly, but not cultured the biofilm;

Response: The biofilm has been cultured to meet the DNA genomic isolation protocol requirements as the extracted DNA quantity should be enough for the identification step and this required enrichment the biofilm by culturing the biofilm. Moreover, culturing of the biofilm was also done in previous studies (ref. 30)

This explanation has been added in the revised manuscript.

P8, line17-19, what the purpose for amplifying the 16S rRNA gene?

Response: 16s rRNA is used for the bacterial strains identification because it is a perfect universal target, and this gene is amplified using PCR to increase the detection sensitivity of the band when running in electrophoresis gel (ref. 31).

This explanation has been added in the revised manuscript.

Results and Discussion

P9, line10 and others, ‘the ambient temperature’ should give a temperature arrange;

Response: The ambient temperature in January in Jeonju, South Korea was with the range of 0 to 5 °C.

This explanation has been added in section “2.3. MFC Construction and operation”, page 7.

P10, section ‘3.2. Aerobic Conditions Effect’, how to maintain the anaerobic conditions? and What the difference with the aerobic condition?

Response: Running the RMFC under anaerobic conditions was performed by purging first the anolyte using nitrogen gas bubbling for 5 min before using in the RMFC and closing the anolyte feeding hole in the anolyte chamber in the assembled MFC. A photo image can be found in the supporting information (Fig. S1) showing the closed inlet hole. On the other hand, aerobic conditions were applied by utilizing the anolyte without purging and opening the anolyte feeding hole; this strategy has been used according to literature (refs. 26 & 27).

This explanation has been added in the revised manuscript, page 7, line 5.

P19, section’ 3.7.2. Bacterial Diversity of RMFC’, confirm all the formats of the microbial names, such as the lactobacillus sp., staphylococcus sp., Staphylococcus sp.....need more work to do; Table 2, Table S2 and S3 also should be corrected in formats.

Response: All microbial names have been checked and corrected as advised.

Reviewer: 2

Comments to the Author(s)

I think that the idea presented in the paper is very original and worth keep exploring in the future.

The only aspect I think should be considered in the paper is an estimation about the energy produced by the Fuel cells. Authors are focused in monitoring voltage produced in the fuel cell, but as the intended use of such a device is to provide power, voltage is not sufficient. The authors measure an I-V curve every now and then, but this is not sufficient as the power delivered by a system with no forced convection will not be constant.

Response: We appreciate the efforts of the reviewer in evaluating the manuscript. Actually, the reviewer could successfully catch the main target of the manuscript.

The power output has been calculated for both summer and winter rainwaters after different days of operation as shown in Fig. 3, in the manuscript. Moreover, two tables (1&2) summarizing the generated powers were added in the revised manuscript.

Also, the lazy metabolism of bacteria may prevent the system from producing the current reported in the I-V in a continues manner. Therefore I think that a discussion and some estimations about energy generated in the cells in such a long period of time is mandatory. I agree that sensors can operated with only few mW but for how long. How big should a FC be to continuously provide power to such sensors?

Response: It is a good comment from the reviewer. First, two tables summarizing the generated powers were added to the revised manuscript. Furthermore, beside the proposed running root described in sec 3.2, this explanation has been added too.

Although, it seems that the generated power from the proposed RMFCs is small, however utilizing the rainwater without inoculation strongly recommends exploiting these cells as power sources in the remote areas as duplicating the power can be achieved by assembling the individual cells in a stack. Moreover, the successful running of the proposed microbial fuel cells under aerobic conditions changes the proposed cells to be a sustainable source of power. Simply, if the constructed cells were left open to the

atmosphere, the cells will be loaded by fresh rainwater which changes the system to be a continuous power source.

Are there any pictures of the fuel cells used in the experiment?

Response: A photo image of the used MFC has been added in the supporting information as Fig. S1.

Can the authors also describe how an operative system in the field would operate?

Would it collect only some rainwater at the beginning of the season, or would the system accept rain through the whole period?

Response: Indeed, this suggested point should be evaluated and examined in the future work to investigate the possibility of constructing a prototype RMFC in the field with the suggested operation mode in section "3.2". Briefly, the proposed RMFC is suggested to work under pseudo continuous mode at the remote areas. In other words, the future design of the RMFC will be based on an open cell configuration having the possibility of filling the cell by fresh rainwater to remove dead microorganisms and activating the biofilm as well as providing nutrients. Moreover, the maintenance of anaerobic conditions in MFC is not easy for the real application³³. Consequently, the performance of the introduced cells was investigated under aerobic conditions". Overall, this study is a proof of the concept for using the rainwater as analyte for MFC, although the operation mode worth to be investigated in the future.

Appendix B

Dear

Subject Editor of Royal Society Open Science Journal

Thank you for accepting our manuscript (ID **RSOS-210996**) titled "**Rainwater-driven Microbial Fuel Cells for Power Generation in the Remote Areas**" for publication in your journal.

The given comments are helpful to strength the manuscript. We would like to inform you that we have modified the manuscript according to the given comments from review: 1 and we would thank both reviewers for their efforts.

We have written the comments in bold phase followed by the responses in normal one. Moreover, in the revised manuscript, you can find the changes in the text in blue color.

We hope our responses cover all the comments.

Thank you for your cooperation

Sincerely yours

Nasser A. M. Barakat

Professor

Minia University

Reviewer: 1

Comments to the Author(s)

Comment: you should know that the microbial community structure of the cultured biofilm is totally different from the real structure of anodic biofilm. You should pay more attention on the formats for the microbial names, such as the strains in 'Table 4 ' should not be in the italic,

Response: from what we found on published articles on MFC area, the microbial names are on italic as shown on the following published articles:

1. Article title “Electrical output of bryophyte microbial fuel cell systems is sufficient to power a radio or an environmental sensor” on “[Royal Society Open Science](https://royalsocietypublishing.org/doi/10.1098/rsos.160249)” journal (<https://royalsocietypublishing.org/doi/10.1098/rsos.160249>)

Review history

Electrical output of bryophyte microbial fuel cell systems is sufficient to power a radio or an environmental sensor

Abstract

Plant microbial fuel cells are a recently developed technology that exploits photosynthesis in vascular plants by harnessing solar energy and generating electrical power. In this study, the model moss species *Physcomitrella patens*, and other environmental samples of mosses, have been used to develop a non-vascular bryophyte microbial fuel cell (bryoMFC). A novel three-dimensional anodic matrix was successfully created and characterized and was further tested in a bryoMFC to determine the capacity of mosses to generate electrical power. The importance of anodophilic microorganisms in the bryoMFC was also determined. It was found that the non-sterile bryoMFCs operated with *P. patens* delivered over an order of magnitude higher peak power output ($2.6 \pm 0.6 \mu\text{W m}^{-2}$) than bryoMFCs kept in near-sterile conditions ($0.2 \pm 0.1 \mu\text{W m}^{-2}$). These results confirm the importance of the microbial populations for delivering electrons to the anode in a bryoMFC. When the bryoMFCs were operated with environmental samples of moss (non-sterile) the peak power output reached $6.7 \pm 0.6 \text{ mW m}^{-2}$. The bryoMFCs operated with environmental samples of moss were able to power a commercial radio receiver or an environmental sensor (LCD desktop weather station).

2. Article title “Effect of fermentation stillage of food waste on bioelectricity production and microbial community structure in microbial fuel cells” on “[Royal Society Open Science](https://royalsocietypublishing.org/doi/10.1098/rsos.180457)” journal (<https://royalsocietypublishing.org/doi/10.1098/rsos.180457>)

Advenella is a type of mesophilic bacterium that can be separated during different composting processes. This bacterium is a new strain of tetracycline-degrading bacteria and can be obtained from pharmaceutical factory wastewater after separation and screening. *Advenella* was demonstrated to degrade tetracycline under suitable conditions of pH 7.0 and 30°C. After 6 days of culture, degradation rate of tetracycline reached 57.8% at an initial concentration of 50 µg ml⁻¹. In our research, *Advenella* occupied the highest proportion among all genera on both anodes and cathodes. Tetracycline possibly exists in food waste and cannot be degraded. As Lee *et al.* discovered, tetracycline antibiotic-resistance gene widely exists in food waste-recycling wastewater [32]. Tetracycline also remains in stillage after fermentation and several times of reflux. Microorganisms in the MFC were screened by tetracycline during operation. The proportion of *Advenella* was extremely high relative to those of other bacteria, and its proportion on the anode was markedly higher than that on the cathode. This result was probably due to the pretreatment of anode materials to be hydrophilic, which benefited growth of microorganisms, whereas cathode materials were pretreated to be hydrophobic. Thus, microbial community structure on the anode differed from that on the cathode. *Moheibacter* was another high-content bacterium on the anode.

Moheibacter is a Gram-negative, non-sliding bacillus that presents bright-yellow, round, smooth and mucoid colonies. Studies showed sensitivity of *Moheibacter* to tetracycline. Thus, *Moheibacter* grew well in the presence of *Advenella*. *Moheibacter* can grow under conditions of 4°C to 33°C, pH 6.0–10.0 and 0% to 3.0% (w/v) NaCl, with optimum growth

3. Article title “A 1.5 µL microbial fuel cell for on-chip bioelectricity generation” on “[Lab on a Chip](https://pubs.rsc.org/en/content/articlehtml/2009/lc/b910586g)” journal (<https://pubs.rsc.org/en/content/articlehtml/2009/lc/b910586g>)

Table 1 Summary of miniturized MFCs in literature

Anode volume	Inoculum	Substrate	Anode material and area	Catholyte	P max (W/m ³)	P max (mW)
7 mL	Geobacter sulfurreducens	acetate	Au (7.8 cm ²)	N/A	N/A	N/A
2.5 mL	Mixed bacterial culture	acetate	carbon cloth (7 cm ²)	air	1010	1800
1.2 mL	Shewanella oneidensis DSP-10	lactate	graphite felt (2 cm ²)	ferricyanide 500 ^ε	3000 ^ε	
			reticulated vitreous carbon (2 cm ²)	140 ^ε	400 ^ε	
16 µL	Saccharomyces cerevisiae	glucose	Au (0.51 cm ²)	ferricyanide 0.5		N/A
10 µL	Shewanella putrefaciens	lactate	Au (0.02 cm ²)	N/A	N/A	N/A
1.5 µL	Shewanella oneidensis MR-1	lactate	Au (0.15 cm ²)	ferricyanide 15.3		1.5

^a Calculation based on cross section of the device or surface area of the anode electrode. ^b Under a poised potential of +300 mV vs.Ag/AgCl. ^c Artificial electron

Therefore, we followed the published format for the microbial names.

Comment: s should be capital in '16s'

Response: the format of 16S has been corrected as advised.